# A Novel Method That Allows SNP Discrimination with 160:1 Ratio for Biosensors Based on DNA-DNA Hybridization

**DOI:** 10.3390/bios11080265

**Published:** 2021-08-06

**Authors:** Satish Balasaheb Nimse, Keum-Soo Song, Shrikant Dashrath Warkad, Taisun Kim

**Affiliations:** 1Department of Chemistry, Institute for Applied Chemistry, Hallym University, Chuncheon 200-702, Korea; satish_nimse@hallym.ac.kr; 2Biometrix Technology, Inc. 202 BioVenture Plaza, Chuncheon 200-161, Korea; kssong@bmtchip.com (K.-S.S.); shrikant@bmtchip.com (S.D.W.)

**Keywords:** biosensors, tuberculosis, DNA-DNA hybridization, single nucleotide polymorphism, signal to background ratio

## Abstract

Highly sensitive (high SBR) and highly specific (high SNP discrimination ratio) DNA hybridization is essential for a biosensor with clinical application. Herein, we propose a method that allows detecting multiple pathogens on a single platform with the SNP discrimination ratios over 160:1 in the dynamic range of 10^1^ to 10^4^ copies per test. The newly developed SWAT method allows achieving highly sensitive and highly specific DNA hybridizations. The detection and discrimination of the MTB and NTM strain in the clinical samples with the SBR and SNP discrimination ratios higher than 160:1 indicate the high clinical applicability of the SWAT.

## 1. Introduction

Biosensors based on concurrent DNA-DNA hybridization for the simultaneous detection of various bacterial strains of the same species have unparallel importance in selecting accurate drug therapy [1,2]. DNA biosensors have become the leading diagnostic technology for detecting and discriminating genotypes of various bacterial and viral strains [3,4]. The sensitivity of the DNA biosensors depends on two factors: (i) the signal to background ratio (SBR) and (ii) the specific to non-specific hybridization ratio, which is also called the single nucleotide polymorphism (SNP) discrimination ratio. Therefore, for any DNA biosensor to be highly accurate and specific in detecting and discriminating against bacterial genotypes, the SBR and SNP discrimination ratio should be sufficiently high [5,6]. However, the reported DNA biosensors suffer from common drawbacks of a very low SBR and SNP discrimination ratio of 10:1 and 5:1, respectively [7,8,9].

The low SBR is related to the immobilization chemistry used for the fabrication of the DNA biosensors. There have been substantial studies on the improvement of the SBR [10,11]. However, achieving a significantly higher SNP discrimination ratio for direct clinical applications is challenging [12]. Therefore, several approaches, such as the longer hybridization time (in hours), use of an elevated hybridization [13,14,15] and washing temperatures [16,17,18,19] close to the temperature of melting (T_m_, °C), and high concentrations of chaotropic agents in hybridization buffers, have been used to increase the specificity [20,21]. However, hybridizations at the probe T_m_ are not always viable; for example, multiplexed DNA biosensors where many different hybridization reactions must progress simultaneously. It is important to note that elevated temperature in order to improve the SNP discrimination ratio results in a lengthy hybridization time (4–16 h) [22]. High temperature also decreases SBR due to the lower hybridization yield of the target DNA with the immobilized probe.

A biosensor must produce final results in less than an hour using a simple experimental protocol and non-astringent conditions (e.g., hybridization and washing temperatures of 25 °C) to be applicable for clinical use. Therefore, to improve the SBR and SNP discrimination ratio, it is essential to achieve a very high signal intensity for the target-specific hybridization compared to the non-specific hybridization and the background signals.

Herein we present an exemplary framework of the mechanism for the highly specific hybridization of the immobilized probes with the Cy5 labeled target DNA (Cy5-ssDNA) obtained by the polymerase chain reaction (PCR). Next, we explain the method for determining the SNP discriminating washing temperature (SWAT) and its application in order to achieve a significantly higher SNP discriminating ratio (>160:1) compared to the reported methods (5:1) [23]. Finally, the application of the developed method, SWAT, for the detection and discrimination of the *Mycobacterium tuberculosis* (MTB) and non-tuberculous mycobacteria (NTM) strains in the clinical samples is presented [24].

## 2. Materials and Methods

### 2.1. Materials

Chemicals were procured from Sigma-Aldrich Chemicals, Korea. All the oligonucleotides, PCR pre-mix, and DNA extraction kits were purchased from Bioneer, Korea. All washing solvents for the substrates are of HPLC grade from SK Chemicals, Seoul, Korea. Ultrapure water (18 M Ω/cm) was obtained from a Milli-Q purification system (Millipore). The 9G DNAChips were obtained from Biometrix Technology Inc., Chuncheon, Korea. The standard samples of *Mycobacterium Tuberculosis* (MTB) and NTM genotypes were obtained from the Korean Institute of Tuberculosis. Mycobacterium organism strains used: *Mycobacterium Tuberculosis* (H37RV); *Mycobacterium Avium* (*M. Avium*); *Mycobacterium Abscessus* (*M. Abscessus*), *Mycobacterium Chelonae* (*M. Chelonae*), and *Mycobacterium Kansasii* (*M. kansasii*).

### 2.2. Composition of Different Solutions 

Hybridization buffer (pH = 7.4) was constituted by mixing 25% formamide, 0.1% Triton X-100, and 6xSSC in double-distilled water. The washing buffer solution A (pH = 7.4) and washing buffer solution B (pH = 7.4) were obtained using 0.1% SDS in 4xSSC and 4xSSC, respectively.

### 2.3. Typical Method for the Preparation of TB-NTM 9G DNAChip

The TB-NTM 9G DNAChips were prepared by spotting the immobilization solution containing oligonucleotide probes Probe1-Probe6 and Probe7-Probe12, respectively, with the microarray, and the spots were arranged to make 6 by 6 pixels on the 9G slides. The microarrayed 9G slides were then kept in the incubator (25 °C, 50% humidity) for 4 h in order to immobilize the oligonucleotides. The slides were then suspended in the blocking buffer solution at 25 °C for 30 min to remove the excess oligonucleotides and deactivate the non-spotted area. Then, the slides were rinsed with washing buffer solutions A and B for 5 min each and then dried with a commercial centrifuge in order to obtain the TB-NTM 9G DNAChips. Before hybridization, the TB-NTM 9G DNAChips were covered with Secure-Seal™ hybridization chambers.

### 2.4. Typical Hybridization and Washing Method

Hybridizations were performed using the Cy5 labeled PCR products containing Cy5ssDNA obtained by asymmetric PCR amplification of the targeted section of the genomic of the MTB and NTM genotypes at 25 °C for 30 min in the commercial incubator. Then, the TB-NTM 9G DNAChips were washed in washing buffer solutions A and B for 2 min each at 35 °C unless otherwise stated, and dried with a commercial centrifuge (1000 rpm). The fluorescence signal of the microarray was measured on ScanArrayLite, and the images were analyzed by Quant Array software (Packard Bioscience).

## 3. Results

In DNA biosensors, the specificity of DNA–DNA hybridization in terms of a high SNP discrimination ratio often depends on several factors, such as (i) a higher hybridization yield for target-specific hybridization than that of non-specific hybridization, (ii) the immobilized probe and Cy5-ssDNA concentrations, (iii) the hybridization and washing temperature, and (iv) the chemical composition of the solvent hybridization buffer. This research aimed to develop a method to determine the washing temperature in order to achieve high SNP discrimination ratios. Hence, the hybridizations were performed at 25 °C, and the washing temperatures were varied from 25 °C, 35 °C, 45 °C, and 55 °C. All other parameters, such as the hybridization buffer (25% formamide, 0.1% Triton X-100, in 6xSSC) and the amount of immobilized probes (6.3 pmol/cm^2^) on the 9G DNAChips were kept constant in all experiments. The use of PCR products obtained with 10^1^ to 10^4^ copies of the genomic DNAs of MTB and various NTM strains allowed determining the robustness of the resented method.

The MTB strain and NTM strains have almost 93–95% sequence homology (see the Appendix A) [25]. Hence, only one PCR primer set was designed to amplify the genomic DNAs of MTB and NTM strains. The single-stranded Cy5 labeled PCR products (Cy5-ssDNA) were obtained by asymmetric PCR amplification using a forward primer (5′-Cy5- CA AGG AGA AGC GCT ACG ACC TGG C-3′) and reverse primer (5′-CCG AAG TGG TCG ATG TCG TC-3′).

The DNA hybridization process depends on the formation of the hydrogen bonding between the complementary nucleotides in two single-stranded DNAs. The DNA hybridization is an exothermic process, and hence it is easy to understand that the increase in the hybridization temperature will significantly decrease the hybridization yield [26]. Therefore, in order to achieve a higher hybridization yield, the 25 °C was chosen as the hybridization temperature. However, as depicted in Scheme 1, the hybridization at a significantly lower temperature than the T_m_ of the immobilized probes results in strong target-specific and non-specific hybridizations, consequently lowering the SNP discrimination ratio.

It is assumed that if the immobilized probe contains one mismatch in its sequence compared to the complementary Cy5-ssDNA, the mismatched nucleotide and two other nucleotides on either side of it do not take part in the hybridization event (Scheme 1a). Thus, based on the nucleotides present in the sequence, the T_m_ of the probe can be reduced by 6 °C (e.g., AAA, ATA), 8 °C (e.g., GAA, CTT), 10 °C (e.g., AGG, CCT), or 12 °C (e.g., GGG, CCC). If the probe contains two mismatches separated by more than three nucleotides in its sequence compared to the complementary Cy5-ssDNA, six nucleotides will not participate in the hybridization (Scheme 1b). Thus, the T_m_ of the probe can be decreased by 12 °C (e.g., AAA and ATA), 14 °C (e.g., AAA and GAA), 16 °C (e.g., GAA and CTT), 18 °C (e.g., GAA and AGG), 20 °C (e.g., AGG and CCT), 22 °C (e.g., AGG and CCC), or 24 °C (e.g., GGG and CCC). However, if the probe contains two consecutive mismatches, only four nucleotides will not participate in the hybridization (Scheme 1c). Thus the T_m_ of the probe can be decreased by 8 °C (e.g., AAAA) to 16 °C (e.g., GGGG). Therefore, to check these assumptions, the 9G DNAChips were obtained by immobilizing the probes according to the previous report [27,28,29,30].

The T_m_ of the probes (Probe1: 5′-TAC CGA CCC ACG CGG GC-3′ (T_m_ = 55 °C); Probe2: 5′-TAC CGG CCC ACG CGG GC-3′ (T_m_ = 57 °C); Probe3: 5′-TAC CGG CCC ACC CGG GC-3′ (T_m_ = 57 °C)) were calculated according to the nearest neighbor method [31,32], and also by the general equation, T_m_ = [4(G + C) + 2(A + T) − 5] °C [33]. The T_m_ of the probes calculated by the nearest neighbor method was identical (±1.5 °C) with the T_m_ calculated by the latter approach (see the Appendix A). We observed the hybridization behaviors by allowing Probe1 and Probe2 to hybridize with the respective complementary Cy5-ssDNA (Figure 1a,b) at 25 °C for 30 min. Then, the 9G DNAChips were washed for 2 min each in washing buffer solutions A (0.1% SDS and 4xSSC) and B (4xSSC) at 25 °C, 35 °C, 45 °C, and 55 °C. All experiments were performed in triplicate.

At a washing temperature of 25 °C, the Cy5-ssDNA complementary to Probe2 showed a specific hybridization with Probe2 indicated by the signal intensity of 60,000 (Figure 1a). In addition, non-specific hybridizations with Probe1 and Probe3 are indicated by the signal intensities of 45,000 and 58,000, respectively. The SNP discrimination ratios for Probe2 to Probe1 and Probe2 to Probe3 were 1.03:1 and 1.33:1, respectively. These SNP discrimination ratios are very low for any clinical application. Therefore, the washing temperature was increased from 25 °C to 55 °C. At 55 °C, the intensities for the non-specific hybridizations decrease drastically compared to the target-specific hybridization. At 55 °C, the complementary Cy5-ssDNA target of Probe2 showed a slight decrease in target-specific hybridization with Probe2, indicated by the signal intensity of 48,000. The non-specific hybridizations with Probe1 and Probe3 were decreased drastically, indicated by the signal intensities of 25,000 and 33,000, respectively. Sequences of Probe1, Probe2, and Probe3 were examined in order to rationalize the reason behind the trends of non-specific hybridizations.

Probe1 and Probe3 have one mismatch in their sequences compared to the probe binding region of the Cy5-ssDNA complementary to Probe2. Based on the explanation of Scheme 1, the T_m_ values of Probe1 and Probe3 for hybridization with the Cy5-ssDNA target complementary to Probe2 decreases from 55 °C and 57 °C to 45 °C and 43 °C, respectively. The T_m_ of Probe1 decreases from 55 °C to 45 °C because three nucleotides (GAC) do not participate in hybridization. Similarly, the T_m_ of Probe3 reduces from 57 °C to 43 °C because three nucleotides (CCC) do not participate in hybridization. The decrease in respective T_m_ values is indicated by the significant reduction in signal intensities of Probe1 and Probe3 upon washing at a higher temperature. Here, the decrease in T_m_ due to mismatches is called T_mdm_ (T_m_ due to mismatch, °C). In similar experiments, the Cy5-ssDNA target complementary to Probe1 showed specific hybridization with Probe1 and non-specific hybridization with Probe2 at 25 °C, indicated by the signal intensities of 60,000, respectively (Figure 1b).

The signal intensity for the non-specific interaction of Probe2 with the Cy5-ssDNA target of Probe1 sharply decreases with the increase in washing temperature (the signal intensity decreases from 60,000 to 18,000). Thus, by washing at 55 °C, the SNP discrimination ratio of Probe1 to Probe2 was only 2.2:1. However, it is interesting that Probe3 did not show any non-specific hybridization with the Cy5-ssDNA target complementary to Probe1. Therefore, even when washing at 25 °C, the SNP discrimination ratio of Probe1 to Probe3 was more than 160:1. These results of the non-specificity of Probe2 and the specificity of Probe3 can be rationalized by the fact that Probe2 has only one mismatch and Probe3 has two mismatches separated by five nucleotides, compared to the probe binding region of the Cy5-ssDNA target complementary to Probe1. Therefore, based on the explanation of Scheme 1, the T_m_ of Probe2 for hybridization with the complementary Cy5-ssDNA target of Probe1 decreases from 57 °C to 45 °C because three nucleotides (GGC) do not participate in hybridization. The sharp decline in signal intensity of Probe2 upon washing at a higher temperature indicates the decrease in T_m_. Similarly, the T_m_ of Probe3 for hybridization with the complementary Cy5-ssDNA target of Probe1 decreases from 57 °C to 33 °C because six nucleotides (CCC and GGC) do not participate in the hybridization process. The reduction in T_m_ is indicated by a lack of hybridization signal for Probe3, even at a washing temperature of 25 °C. Therefore, from the result of the hybridization behavior of Probe3 (T_m_ of 57 °C and T_mdm_ of 33 °C), it was concluded that washing at a temperature higher than T_mdm_-10 °C could eliminate the non-specific hybridization and allow for the SNP discrimination ratio to be higher than 160:1.

To further understand the concept, new probes were designed containing one artificial mutation (see the Appendix A). Probe1 was changed to Probe4 (5′-TAC CGA CCC ACG TGG GC-3′ (T_m_ = 53 °C)) by adding the artificial mutation at the 5th position from the 3′ end. Probe2 was changed to Probe5 (5′-TAC CGG CCT ACG CGG GC-3′ (T_m_ = 55 °C)) by adding the artificial mutation at the 9th position from the 3′ end. Probe3 was changed to Probe6 (5′-TAC CGG CCC ACC TGG GC-3′ (T_m_ = 55 °C)) by adding the artificial mutation at the 5th position from the 3′ end.

As demonstrated in Figure 1c at washing temperatures of 25 °C to 45 °C, the complementary Cy5-ssDNA target of Probe2 showed an excellent specific hybridization with Probe5, indicated by the signal intensity of 60,000. The non-specific hybridizations with Probe6 at washing temperatures of 25 °C were suggested by the signal intensity of 10,000, which was vanished upon washing at higher temperatures. Therefore, at 25 °C washing temperatures, the SNP discrimination ratio of Probe4 to Probe6 was 60:1, which was further improved to >160:1 by washing at 35 °C. Interestingly, Probe4 did not show any non-specific hybridizations with the Cy5-ssDNA target complementary to Probe2.

Similar results were also observed for the hybridization of the Cy5-ssDNA target complementary to Probe1 with Probe4, Probe5, and Probe6 (Figure 1c). Probe4 demonstrated a highly specific hybridization, indicated by the signal intensities of 60,000 upon washing at 25 °C and 35 °C, respectively. However, upon washing at higher temperatures (45 °C and 55 °C), the hybridization signals decreased to 47,000 and 34,000, respectively. Interestingly, Probe5 and Probe6 showed no non-specific hybridization with the Cy5-ssDNA target complementary to Probe1, even at a washing temperature of 25 °C. The SNP discrimination ratios of Probe4 to Probe5 and Probe4 to Probe6 were higher than 160:1.

The hybridization patterns of Cy5-ssDNA targets complementary to Probe2 and Probe1 with Probe4, Probe5, and Probe6 can be rationalized by comparing their sequences. Probe6 has two consecutive mismatches compared to the probe binding region of the Cy5-ssDNA target complementary to Probe2. Therefore, the T_m_ of Probe6 for the hybridization with complementary Cy5-ssDNA target of Probe2 decreases from 55 °C to 41 °C because four nucleotides (CC TG) do not participate in hybridization. Therefore, according to the T_mdm_-10, the SNP-discriminating washing temperature for this probe should be higher at 31 °C. It is essential to note that the non-specific hybridization was observed as anticipated upon washing at 25 °C. The non-specific hybridization was eventually disappeared by hybridization at 35 °C, leading to a SNP discrimination ratio higher than 160:1.

Similarly, Probe4 has two mismatches separated by more than four nucleotides, compared to the probe binding region of the Cy5-ssDNA target complementary to Probe2. The T_m_ of Probe4 for the hybridization with Cy5-ssDNA target complementary to Probe2 decreases from 53 °C to 23 °C because six nucleotides (GAC and GTG) do not participate in hybridization. Therefore, according to the T_mdm_-10, the SNP discriminating washing temperature for this probe should be higher than 23 °C. Thus, Probe4 does not show non-specific hybridization with the complementary Cy5-ssDNA target of Probe2 at a washing temperature of 25 °C, which is higher than T_mdm_-10.

The data presented in Figure 1 indicate that, due to one mismatch in the sequences of the ssDNAs, the double helix is distorted during hybridization in such a way that three nucleotides (mismatched nucleotide and one nucleotide on either side) do not participate in the hybridization (Scheme 1a). Therefore, if over three nucleotides separate the two mutations in the probes, approximately six nucleotides (S = 6) do not participate in the hybridization with the target ssDNA (Scheme 1b). Similarly, if the two mutations are consecutive, four nucleotides (C = 4) in the probe will not participate in the hybridization with the target ssDNA (Scheme 1c). Therefore, the T_mdm_ for any given sequence can be obtained by the following equations,
T_mdm_ = S × 3, °C(1)
T_mdm_ = C × 3, °C(2)
where S = 6 (if there are two mutations in the probe separated by more than three nucleotides) and C = 4 (if there are two consecutive mutations in the probe). Therefore, from the results presented in Figure 1 and by using Equations (1) and (2), the equation for the determination of the SNP-discriminating washing temperature (SWAT) can be obtained by the following equations,
SWAT = (S × 3) + 10, °C(3)
SWAT = (C × 3) + 20, °C(4)

Therefore, the non-specific hybridizations can be eliminated by implementing washing temperatures higher than the SWAT after hybridization at 25 °C. The TB-NTM 9G DNAChips were used to investigate the applicability of the SWAT for the detection and discrimination of the MTB and NTM strains in the clinical samples. The TB-NTM 9G DNAChips were obtained by the immobilization of Probe7 (*MTB*), Probe8 (*M. Chelonae*), Probe9 (*M. Avium*), Probe10 (*M. Marinum*), Probe11 (*M. Abscessus)*, Probe12 (*M. Kansasii*), Probe13 (PC), Probe14 (PCR), and Probe15 (HC) on the AMCA slides, following the earlier report. The immobilized probes were allowed to hybridize at 25 °C with the Cy5-ssDNA obtained by the asymmetric PCR amplification of the targeted genomic DNA of the MTB and four NMT strains. The DNA chips were washed at 35 °C after hybridization in order to keep the experimental protocol simple. The obtained results are depicted in Figure 2.

The results in Figure 2 demonstrate that, after hybridization at 25 °C followed by washing at a temperature (25 °C for Probe7–12) slightly higher than SWAT, the highly sensitive and highly specific detection can be easily achieved. Figure 2 also demonstrates that the SNP discrimination ratio in most cases was over 160:1. Therefore, these results confirm the clinical applicability of the SWAT on the TB-NTM 9G DNAChips and assure the boost in the clinical study for the detection and discrimination of MTB and NTM strains for accurate drug therapy.

In order to determine the robustness of the presented method in the wide concentration range, the probes immobilized on the TB-NTM 9G DNAChips were allowed to hybridize at 25 °C with the PCR products obtained from the 10^1^ to 10^4^ copies of the genomic DNA of the MTB and NTM strains, followed by the washing at 35 °C. The obtained results (see the Appendix A) demonstrate the SWAT’s applicability in the highly sensitive and highly specific detection and discrimination of the MTB and NTM strains in the dynamic range of 10^1^ to 10^4^ copies per test.

The clinical samples comprising the MTB and NTM strains allowed us to evaluate the clinical applicability of the proposed method (see the Appendix A). The results demonstrate that the proposed method allows for the correct detection and discrimination of the MTB and NTM strains in the clinical samples. The SNP discrimination ratio for detecting and discriminating the MTB and NTM strains in the clinical samples was higher than 160:1. Therefore these results confirm the clinical applicability of SWAT.

## 4. Discussion

The methods for the development of microarrays to detect infectious agents have been extensively studied. The probe design is a central science in successfully applying microarrays, since it directly correlates with the SNP discrimination efficiency. The non-specific hybridization leading to poor SNP discrimination ratios (5:1) remains a significant disadvantage of microarray-based methods. Several studies report on improving the SNP discrimination ratio by increasing the hybridization and washing temperatures (45–95 °C) and lengthy processes (4–16 h) [34]. Higher temperatures in hybridization and washing steps improve the SNP discrimination ratio but at the expense of the sensitivity of the assay [35]. Another aspect is that higher temperatures impart complications in the detection process, which otherwise would be simple if conducted at room temperatures. The reproducibly of the biosensor assay platforms is another crucial aspect for their successful implementation in clinical practice.

In this regard, we have developed a method for determining the SNP-discriminating washing temperature (SWAT). The SWAT allows for the design of probes that do not show non-specific interactions, even at room temperatures, unlike the reported methods where they need to use high temperatures. The SWAT allows determining the washing temperature and helps to design the probe sequence such that the SBR and SNP-discriminating ratio is >160:1, compared to the reported methods (5:1). The present method is applied to detecting and discriminating MTB and well-known NTM strains responsible for infectious diseases. The results shown in Appendix A indicate the high reproducibility of the presented method. The detection of MTB and NTM strains in the clinical samples with 100% accuracy, and the application of SWAT in the development of the probes for more than one strain of NTM, also assures reproducibility of the proposed method. The results of the clinical study promise the applicability of the presented method in clinical settings.

## 5. Conclusions

In conclusion, the highly sensitive and specific hybridization of the target DNA of diagnostic importance with probes is an essential criterion for the clinical applicability of diagnostic methods based on DNA hybridizations. In this communication, we have presented an exemplary framework for the highly specific DNA hybridizations. The newly developed SWAT method allows achieving highly sensitive and highly specific DNA hybridizations with the hybridization and washing steps conducted at 25 °C. The detection and discrimination of the MTB and NTM strain in the clinical samples with the SBR and SNP discrimination ratios higher than 160:1 indicate the high clinical applicability of the SWAT. In turn, SWAT can boost the ongoing efforts to identify the MTB and respective NTM infections for accurate drug therapy.

## Data Availability

The data presented in this study are available on request from the corresponding author.

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
