# Peer review of "A Novel Method That Allows SNP Discrimination with 160:1 Ratio for Biosensors Based on DNA-DNA Hybridization"

_biosensors, 2021, doi:10.3390/bios11080265_

Round 1

Reviewer 1 Report

Authors present in this work a new method for determining “SNP discriminating washing temperature (SWAT)”, which demonstrates the possibility to design probes that do not show non-specific interactions even at room temperatures, and its clinical applicability to discriminate MTB and NTM strains in clinical samples with SBR and SNP discrimination ratios higher than 160:1.

The results presented are of both scientific and clinical interest and the manuscript is well presented and worked. Thus, I recommend its publication in this journal after addressing just a couple of issues:

- The authors should briefly discuss the reproducibility of the developed method.

- Have the authors evaluated the influence of washing time?

Author Response

Reviewer-1

Comments and Suggestions for Authors:

Authors present in this work a new method for determining “SNP discriminating washing temperature (SWAT)”, which demonstrates the possibility to design probes that do not show non-specific interactions even at room temperatures, and its clinical applicability to discriminate MTB and NTM strains in clinical samples with SBR and SNP discrimination ratios higher than 160:1. The results presented are of both scientific and clinical interest and the manuscript is well presented and worked. Thus, I recommend its publication in this journal after addressing just a couple of issues:

We thank the honorable reviewer for the valuable comments. We have modified the manuscript according the suggestions. The changes in the manuscript are indicated by the purple color text. And the correction of grammatical errors are indicated by the text highlighted with yellow color.

Comment-1: The authors should briefly discuss the reproducibility of the developed method.

Answer: We have added the following text to the discussion section of the manuscript.

 Line no 245-246, “The reproducibly of the biosensor assay platforms is another crucial aspect for their successful implementation in clinical practice”.

Line no 252-254, “The results shown in Figure S5 indicate the high reproducibility of the presented method. The detection of MTB and NTM strains in the clinical samples with 100% accuracy and the application of SWAT in the development of the probes for more than one strain of NTM also assures reproducibility of the proposed method.”

Comment-1: Have the authors evaluated the influence of washing time?

Answer:  The effect of washing time was not studied. The shorter turn-around time for the diagnostic assay based on DNA chips is crucial for their clinical applicability. Therefore, the washing time of 2 min with washing buffer A and 2min with washing buffer B were used as it requires a total of 4min washing time. Besides, the PCR products that are physisorbed on the surface and be washed at 25 0C using washing buffers A and B. Hence, we did not study the effect of washing time.

Reviewer 2 Report

The authors address an important problem in biomedical engineering: The development of methods for SNP detection that have high signal-to-background ratio. However, the presentation of the content of the manuscript is not clear. Some of the concerns with the manuscript follows:

Major issues:

  1. The authors should broaden the literature review to include previous studies in which the washing temperature used was higher than the hybridization temperature.
  2. The parameter Tm, presented for the first time in the line 105, is not introduced in the text.
  3. The parameter Tm is a temperature, but it lacks the unit (Celsius?) in many instances, including in the lines 121, 122, 168, 169.
  4. The authors pointed out in the lines 91 and 92 that the goal of the study is to optimize the SNP discrimination ratio on the washing temperature. However, in the line 135, the authors indicated that the hybridization temperature was being varied in the study.
  5. In the line 143, the authors state that the Tm of the Probe 3 decreases from 57 C to 43 C. However, on the line 155, the authors state that the Tm of the Probe 3 decreases from 55 C to 43 C. The authors need to be consistent on that information.
  6. No figure was cited in the discussions in the paragraph from the lines 145 to 148. Moreover, the numerical values provided in those lines (Tm of the Probe 3 decreases from 55 C to 43 C) are not consistent with the figures. Were they obtained from the figures? If that was not the case, an explanation needs to be provided for the source of those numbers.
  7. No figure was cited in the discussions in the paragraph in the lines 161 to 166.
  8. In the lines 151 and 152, the authors stated: “The non-specific hybridization of Probe2 with the Cy5-ssDNA target of Probe1 sharply decreases by the increase in washing temperature.” The hybridization is not affected by the washing temperature. Instead, a proper choice of the washing temperature can break apart non-specific hybridization.
  9. In the lines 154 and 155: The authors stated: “Therefore, even washing at 25 ºC, 154 the SNP discrimination ratio of Probe1 to Probe 3 was more than 160:1.” This was the signal-to-background ratio provided by the authors in the title. However, no optimization in the washing temperature was needed to achieve that number. The authors need to explain why previous work using the same hybridization temperature did not achieve the 160:1 signal-to-background ratio.

Minor issue:

  1. Carry out a thorough review of the presentation in English. For example, the paragraph starting in the line 120 should hot have started with the word “Thus”, since that is a new paragraph. Another instance is the word “high” in line 89. It should have been “higher”.

Author Response

Reviewer-2

Comments and Suggestions for Authors

The authors address an important problem in biomedical engineering: The development of methods for SNP detection that have high signal-to-background ratio. However, the presentation of the content of the manuscript is not clear. Some of the concerns with the manuscript follows:

We thank the honorable reviewer for the valuable comments. We have modified the manuscript according the suggestions. The changes in the manuscript are indicated by the purple color text. And the correction of grammatical errors are indicated by the text highlighted with yellow color.

 Major issues:

  1. The authors should broaden the literature review to include previous studies in which the washing temperature used was higher than the hybridization temperature.

Answer:  We have added the additional references 15-18 as suggested by honorable reviewer.

  1. The parameter Tm, presented for the first time in the line 105, is not introduced in the text.

Answer: The meaning of parameter Tm is added to the text.

Line 37, (close to the temperature of melting (Tm, °C)), 105-107, “However, as depicted in Scheme 1, the hybridization at a significantly lower temperature than the Tm of the immobilized probes results in strong target-specific and non-specific hybridizations, consequently lowering the SNP discrimination ratio.”

  1. The parameter Tm is a temperature, but it lacks the unit (Celsius?) in many instances, including in the lines 121, 122, 168, 169.

Answer:  We thank the honorable reviewer to bring this point into our notice. We have added the “°C” unit where necessary including the lines 121, 122, 168, 169.

  1. The authors pointed out in the lines 91 and 92 that the goal of the study is to optimize the SNP discrimination ratio on the washing temperature. However, in the line 135, the authors indicated that the hybridization temperature was being varied in the study.

Answer: It is true that instead of writing “washing temperature”, "hybridization temperature” was written. We have fixed the sentence as following, “Therefore, the washing temperature was increased from 25 °C to 55 °C.”

  1. In the line 143, the authors state that the Tm of the Probe 3 decreases from 57 C to 43 C. However, on the line 155, the authors state that the Tm of the Probe 3 decreases from 55 C to 43 C. The authors need to be consistent on that information.

Answer:  We have corrected the statement as following both in line 143 and 160-161. " Similarly, the Tm of Probe3 decreases from 57 °C to 43 °C because three nucleotides (CCC) do not participate in hybridization.”

  1. No figure was cited in the discussions in the paragraph from the lines 145 to 148. Moreover, the numerical values provided in those lines (Tm of the Probe 3 decreases from 55 C to 43 C) are not consistent with the figures. Were they obtained from the figures? If that was not the case, an explanation needs to be provided for the source of those numbers.

Answer:  We have provided the information to recalculate the Tm of probes with one or two nucleotide differences in the sequence of probes in the explanation of Scheme 1. The decrease in the Tm of the probes were calculated based on that information.

We have modified the sentence 141-148 as " Based on the explanation of Scheme 1, the Tm values of Probe1, Probe3 for hybridization with Cy5-ssDNA target complementary to the Probe2 decreases from 55 °C, 57 °C to 45 °C, 43 °C, respectively. The Tm of Probe1 decreases from 55 °C to 45 °C because three nucleotides (GAC) do not participate in hybridization. Similarly, the Tm of Probe3 reduces from 57 °C to 43 °C because three nucleotides (CCC) do not participate in hybridization. The decrease in respective Tm values are indicated by the significant reduction in signal intensities of Probe1 and Probe3 upon washing at a higher temperature. Here, the decrease in Tm due to mismatches is called Tmdm (Tm due to mismatch, °C).

We have modified the sentence 158 in similar way.

  1. No figure was cited in the discussions in the paragraph in the lines 161 to 166.

Answer:  We have modified the sentence 158-166 in similar way as mentioned in the answer of comment 6.  

  1. In the lines 151 and 152, the authors stated: “The non-specific hybridization of Probe2 with the Cy5-ssDNA target of Probe1 sharply decreases by the increase in washing temperature.” The hybridization is not affected by the washing temperature. Instead, a proper choice of the washing temperature can break apart non-specific hybridization.

Answer:  we agree with the honorable reviewer. We have modified the sentence as follows, “The signal intensity for the non-specific interaction of Probe2 with the Cy5-ssDNA target of Probe1 sharply decreases by the increase in washing temperature (signal intensity decreases from 60000 to 18000).”

  1. In the lines 154 and 155: The authors stated: “Therefore, even washing at 25 ºC, 154 the SNP discrimination ratio of Probe1 to Probe 3 was more than 160:1.” This was the signal-to-background ratio provided by the authors in the title. However, no optimization in the washing temperature was needed to achieve that number. The authors need to explain why previous work using the same hybridization temperature did not achieve the 160:1 signal-to-background ratio.

Answer:  The SNP discrimination ratio of 160:1 is cited in the title of this article. We would also like to point out that the SBR and SNP discrimination ratio were found to be 160:1. The background signal in spot area for possible non-specific interactions were around 350 ~ 400 (black color). Whereas, the signal for the specific-hybridization of the probes with their complementary Cy5-ssDNA targets were about 62000 ~ 65000 (white color).

                We would also like to bring into notice that the presented method demonstrates the optimization of the probe washing temperatures < 25 °C. It is believed that the readers can use the presented method to achieve the SBR and SNP discrimination ratio of > 160:1 using the hybridization and washing temperatures of 25 °C.

 Minor issue:

  1. Carry out a thorough review of the presentation in English. For example, the paragraph starting in the line 120 should hot have started with the word “Thus”, since that is a new paragraph. Another instance is the word “high” in line 89. It should have been “higher”.

Answer: We have worked on the grammatical mistakes in the manuscript and made the necessary correction. The grammatical corrections are indicated by highlighting the text with yellow color.

Line no 89, high -> higher; line 120, “Thus” ->  “Therefore” and the line is merged with the previous paragraph.

Round 2

Reviewer 2 Report

The reviewer thanks the authors for making changes to the manuscript following recommendations during the first review cycle.  I have one concern regarding the revised version:

On the line 52, the authors indicate that the SWAT method that they proposed achieved an SNP discriminating ration of 160:1, while other studies (in which the hybridizing and the washing temperature was fixed) reported only 4:1. The corresponding citation should be included in the manuscript. On the line 31, the authors also indicated that other studies using the same temperature for the hybridizing and the washing achieved SNP discriminating ratios of only 10:1 and 5:1 (references 7 and 8), which are quite low. The authors mentioned on the line 176 that they achieved a discriminating ratio of 60:1 when the washing temperature was the same as the washing temperature (25 C). The authors need to explain why they obtained a discriminating ratio 60:1 when the washing and the hybridizing temperatures were the same while others obtained a much lower discriminating ratio (4:1, 10:1, and 5:1). To make a fair comparison, the authors should compare their results with the SWAT method against those without using SWAT for the same target DNA. In other words, the improvement of SWAT is from 60:1 to 160:1 discriminating ratio. The authors need to provide stronger arguments to justify an improvement on the discriminating ratio from 4:1 to 160:1, since those are likely not applied to the same target DNA.

Author Response

Reviewer-2 (Round 2)

Comments and Suggestions for Authors

The reviewer thanks the authors for making changes to the manuscript following recommendations during the first review cycle.  I have one concern regarding the revised version:

Q1. On the line 52, the authors indicate that the SWAT method that they proposed achieved an SNP discriminating ration of 160:1, while other studies (in which the hybridizing and the washing temperature was fixed) reported only 4:1. The corresponding citation should be included in the manuscript.

Answer: Sentence 52 is corrected to “compared to the reported methods (5:1) [23].Reference 23 is added to the manuscript. We have also added reference 9 (Gynecol Oncol. 2005 Sep;98(3):369-75. doi: 10.1016/j.ygyno.2005.04.044.) to the manuscript. As can be seen from the following figure (reference 9) that the background intensities are around (8000 ~1000) and spot intensities are around (32,000 to 40,000), indicating an SBR of 4:1.

Gynecol Oncol. 2005 Sep;98(3):369-75. doi: 10.1016/j.ygyno.2005.04.044.

Q.2 On the line 31, the authors also indicated that other studies using the same temperature for the hybridizing and the washing achieved SNP discriminating ratios of only 10:1 and 5:1 (references 7 and 8), which are quite low. The authors mentioned on the line 176 that they achieved a discriminating ratio of 60:1 when the washing temperature was the same as the washing temperature (25 oC). The authors need to explain why they obtained a discriminating ratio 60:1 when the washing and the hybridizing temperatures were the same while others obtained a much lower discriminating ratio (4:1, 10:1, and 5:1).

Answer: We understand the concerns of the honorable reviewer that when reported methods used the same hybridization and washing temperatures achieved the discrimination ratio of 5:1. In this study, [Line no. 131-134] the SNP discrimination ratios for Probe2 to Probe1 and Probe2 to Probe3 were 1.03:1 and 1.33:1, respectively (hybridization and washing temperatures of 25 oC.).

              [Line 173-176] The non-specific hybridizations with Probe6 at washing temperatures of 25 °C were suggested by the signal intensity of 10000, which was vanished up on washing at higher temperatures. Therefore, at 25 °C washing temperatures, the SNP discrimination ratio of Probe4 to Probe6 was 60:1.

It is important to note that the reported methods used higher hybridization and washing temperatures to achieve some level of discrimination. Here, it is essential to consider the Tm of the probe. By definition, the Tm of the probe indicates only 50% hybridization, which means hybridization and washing at higher temperatures will allow only a few percent of specific hybridization (low signal intensity of the target probe). The effect of temperature varies drastically on the non-specific hybridizations depending on the substrate used for probe immobilization, immobilization chemistry, blocking reagents, etc. Therefore, the use of higher temperatures in hybridization and washing steps improves the SNP discrimination ratio but at the expense of sensitivity of the assay.

In the present method, we have mainly focused on the Tm of the probes. It is important to note that the SWAT method allows a discrimination ratio of 160:1 even using the hybridization and washing temperatures of 25 oC. The reason is that when designing the probes using the SWAT method, the achieved Tm of the probes are < 25 oC.

For example, Tm of Probe4 for the hybridization with Cy5-ssDNA target complementary to the Probe2 decreases from 53 °C to 23 °C because six nucleotides (GAC and GTG) do not participate in hybridization. Therefore, hybridization and the washing steps at 25 oC allow 90 ~ 95 % hybridization indicated by the signal saturation (white spots). Thus, using SWAT with efficient probes guarantees the improvement in the discrimination ratio from 1.03:1 < 60:1 < 160:1.

Q3. To make a fair comparison, the authors should compare their results with the SWAT method against those without using SWAT for the same target DNA. In other words, the improvement of SWAT is from 60:1 to 160:1 discriminating ratio. The authors need to provide stronger arguments to justify an improvement on the discriminating ratio from 4:1 to 160:1, since those are likely not applied to the same target DNA.

Answer: We agree with the honorable reviewer that we have already demonstrated the data for with and without application of SWAT indicated by the improvement of discrimination ratio from 1.03:1 to 160:1. We believe that the SWAT method presented here can be widely used irrespective of the immobilization chemistry or substrate used for immobilization. The researchers would need to design the probes according to the explanation for SWAT. The hybridization and washing temperatures can be set to 25 oC for proper choice of probes as exemplified for detection and discrimination of MTB and NTM strains in the manuscript.

We agree that the target DNA used in this research are different from those in the published articles. However, the improvement of the discrimination ratio from 1.03:1 < 60:1 < 160:1 is highly significant for improving the clinical sensitivity and specificity of the developed assay. Therefore, we believe that the method presented here will help the researchers in the field to design the probes according to the SWAT method to achieve a higher discrimination ratio.
